# The RUNX/CBFβ Complex in Breast Cancer: A Conundrum of Context

**DOI:** 10.3390/cells12040641

**Published:** 2023-02-16

**Authors:** Adiba S. Khan, Kirsteen J. Campbell, Ewan R. Cameron, Karen Blyth

**Affiliations:** 1Cancer Research UK Beatson Institute, Garscube Estate, Switchback Rd, Glasgow G61 1BD, UK; a.khan@beatson.gla.ac.uk (A.S.K.); k.campbell@beatson.gla.ac.uk (K.J.C.); 2School of Cancer Sciences, College of Medical, Veterinary and Life Sciences, University of Glasgow, Glasgow G61 1QH, UK; 3School of Biodiversity One Health & Veterinary Medicine, College of Medical, Veterinary and Life Sciences, University of Glasgow, Glasgow G61 1QH, UK; ewan.cameron@glasgow.ac.uk

**Keywords:** breast cancer, RUNX1, CBFβ, RUNX2, RUNX3, estrogen receptor (ER), mammary, metastasis

## Abstract

Dissecting and identifying the major actors and pathways in the genesis, progression and aggressive advancement of breast cancer is challenging, in part because neoplasms arising in this tissue represent distinct diseases and in part because the tumors themselves evolve. This review attempts to illustrate the complexity of this mutational landscape as it pertains to the *RUNX* genes and their transcription co-factor CBFβ. Large-scale genomic studies that characterize genetic alterations across a disease subtype are a useful starting point and as such have identified recurring alterations in *CBFB* and in the *RUNX* genes (particularly *RUNX1*). Intriguingly, the functional output of these mutations is often context dependent with regards to the estrogen receptor (ER) status of the breast cancer. Therefore, such studies need to be integrated with an in-depth understanding of both the normal and corrupted function in mammary cells to begin to tease out how loss or gain of function can alter the cell phenotype and contribute to disease progression. We review how alterations to RUNX/CBFβ function contextually ascribe to breast cancer subtypes and discuss how the in vitro analyses and mouse model systems have contributed to our current understanding of these proteins in the pathogenesis of this complex set of diseases.

## 1. Introduction

Breast cancer is one of the leading causes of cancer-related deaths worldwide [1]. Affecting both women and men, the incidence rate of this malignancy has increased by 20% over the last two decades and is predicted to rise further [1].

Breast cancer is characteristically divided into different subtypes and clinically managed based on the expression of the estrogen receptor (ER), progesterone receptor (PR) and human epidermal growth factor receptor 2 (HER2). Luminal breast cancer, which is estrogen receptor positive (ER+) and can also express HER2, comprises about 75% of the cases diagnosed [2,3,4]. While treatment with selective ER modulators (SERMs) such as Tamoxifen; ER degraders (SERDs); and aromatase inhibitors (AI) have successfully improved disease prognosis in this subtype, approximately 40% of patients acquire resistance to these endocrine therapies and succumb to metastatic disease [5,6,7,8]. Furthermore, there is a lack of targeted treatment for the 15% of patients with triple negative, TN (ER−/PR−/HER2−), breast cancer. Thus, the need to discover novel targets and improve current treatment options is of utmost importance. In order to do so, it is crucial to understand the molecular and genetic features involved in the pathogenesis of this malignant disease.

Breast cancer is in part a genetic disease in which the alterations in gene expression patterns and the accumulation of mutations in both tumor suppressor genes and oncogenes drive the transformation of normal breast cells towards malignancy [9]. Around 1600 driver mutations have been reported in 93 breast cancer genes, such as common alterations in *TP53, PIK3CA, PTEN, MYC* and *GATA3* [10,11]. Interestingly, The Cancer Genome Atlas (TCGA) PanCancer Atlas [12,13,14,15,16,17,18,19,20,21] shows that a significant percentage of all breast cancer cases are linked to alterations in the genes encoding the RUNX/CBFβ complex [22,23]. This is also supported by data from the METABRIC study, where 14% of primary breast cancer samples were found to harbor alterations in *CBFB*, while 11%, 5% and 4% of patient samples possessed various categories of genetic alterations in *RUNX1*, *RUNX2* and *RUNX3*, respectively [24] (Figure 1A). Of particular note is the different spectrum of genetic alterations, not only between each of the *RUNX* genes, but also the skewed distribution of these alterations across ER positive versus ER negative disease (Figure 1B).

## 2. The RUNX/CBFβ Transcription Complex

The Runt-related transcription factor (RUNX) family, historically known as the polyoma enhancer-binding protein 2 α (PEBP2α), acute myeloid leukemia (AML) and core binding factor α (CBFα) family of proteins [25], comprises three members: RUNX1, RUNX2 and RUNX3. These factors operate as part of a heterodimeric core binding factor (CBF) complex with their obligate partner—core binding factor-beta (CBFβ) [26,27,28,29]. RUNX1, found to undergo frequent chromosomal translocations in AML patients, is critical in the development, differentiation and homeostasis of hematopoietic stem cells (HSC) [30,31]. RUNX2 is a master regulator in bone development, particularly in the differentiation of osteoblasts [32,33]. RUNX3, expressed in a range of tissues, has an essential role in the differentiation of dorsal root ganglion neurons [34,35] and immune cell regulation [36]. DNA binding affinity and transcriptional activity of these three key players across multiple tissue types are heavily dependent on their transcriptional co-factor: CBFβ [29,30,37]. Together the RUNX/CBFβ complex regulates the transcription of numerous genes involved in cell proliferation, differentiation and survival [38,39].

A key feature common to all three RUNX proteins is the presence of a highly conserved domain comprised of 128 amino acids—the Runt-homology domain (RHD), named after the *Drosophila* runt [40]. This region was found to support two functions: allowing the RUNX proteins to bind to DNA, and enabling heterodimerization to CBFβ [41] (Figure 2). CBFβ is a ubiquitously expressed protein, encoded in mammals by the 50 kb *CBFB* gene [42,43]. CBFβ is a non-DNA binding protein that lacks a nuclear localization signal [44,45] and resides in the cytoplasm where it can bind to the RHD on RUNX proteins [46,47]. While binding to RUNX proteins allows CBFβ to be shuttled into the nucleus (Figure 2), it repays the favor by allosterically stabilizing the point of contact between the RHD and DNA [41], thereby improving the DNA-binding affinity of RUNX proteins by 40-fold [48,49]. Furthermore, binding to CBFβ can protect RUNX proteins, as shown in the case of RUNX1 where heterodimerization with CBFβ prevents its degradation [50]. Indeed, RUNX1 levels are barely detected in *Cbfb*-knockout mice, and CBFβ has been shown to increase the half-life of this protein by preventing proteolysis of RUNX1 via ubiquitination [50].

Inside the nucleus, the RHD is used to modulate transcription of target genes through interactions with specific promoter and enhancer elements [51]. The RHD recognizes and binds the 5′-TGYGGT-3′ consensus sequence [52]—or its inverse complement 5′-R/TAACCRCA-3′—in putative RUNX target promoters (Y = C or T, R = G or A) [39,51]. Cofactors, such as C/EBP, MYB, JUND and ETS, have promoter sites in close proximity to the RHD binding elements and therefore help regulate transcription through direct interaction with DNA. Other coactivators including ALYREF, Yes-associated protein 1 (YAP1) and EP300/CREBBP activate transcription initiation through histone acetylation; direct acetylation of RUNX proteins [53,54], or by recruiting the transcription initiation complex [55]. Direct or indirect interactions with co-repressors—such as Groucho/Transducin-like Enhancer Protein 1 (TLE) proteins, mSin3A, histone de-acetylases (HDACs), nuclear hormone co-receptor (NCOR1) and silencing mediator of retinoid and thyroid hormones (SMRT)—are also used by the RUNX/CBFβ complex to negatively regulate transcription [39,55]. 

Transcription regulation by RUNX/CBFβ is additionally impacted by post-translational modifications, as reviewed by Chuang et al. [56]. Activity of RUNX proteins is dynamically controlled by phosphorylation (via kinases such as cyclin-Cdks, ERK, HIPK2 and PIM1); acetylation (by EP300 and/or in response to BMP2 signaling); methylation (through the PRMT1 methyltransferase); and ubiquitin mediated proteolysis [53,54,57,58,59,60,61,62,63,64,65,66]. Further regulation of RUNX/CBFβ is achieved through chromatin modifications. MOZ and MORF are two acetyltransferases that directly interact and stimulate RUNX function [67,68]. An active chromatin has been associated with the collaboration of RUNX/CBFβ [56] with the chromatin modelling complex SWI/SNF that is formed by SMARC proteins (e.g., SMARCA1/SNF2L1, SMARCA4/BRG1) [69,70]. We should also bear in mind that there is evidence of autoregulation and cross-regulation of the RUNX proteins [71,72,73]. Indeed, the mutually exclusive expression of RUNX1 and RUNX3 in B-lymphoma cells was specifically due to RUNX3 acting on the RUNX binding site within the P1 promoter to repress RUNX1 expression [73].

This comprehensive regulation of, and by RUNX/CBFβ is imperative in controlling cellular pathways in both development and disease (Figure 2C). Indeed, loss of either components of the CBF complex has lethal effects on normal development. Homozygous deletions of *Runx1* or *Cbfb* in vivo prevent hematopoietic development, induce central nervous system hemorrhaging and respiratory issues and ultimately lead to death of embryos within 12–14 days post conception [45,74,75]. Mice with full body deletion of *Runx2* succumb to death from malformations in bone development with a distinct phenotype of concaved rib cages that leads to severe respiratory defects [76,77]. CBFβ, as the co-factor for RUNX2, was also shown to be an essential player in osteogenesis [78,79]. Additionally, RUNX3/CBFβ is a critical regulator of neuronal development alongside mediating the development and function of bone, blood and immune cells [34,35,80,81,82,83,84]. 

Noting how critical RUNX and CBFβ are in normal homeostasis and development, it is not surprising that these genes are frequently altered and mutated in a multitude of cancers [39,85,86,87,88,89]. A compendium of manuscripts commissioned in this special issue of *Cells* comprehensively explores various facets of RUNX/CBFβ in cancer, and so, here we will provide an overview specifically on the *RUNX* genes and CBFβ co-factor in breast cancer where intriguingly mutations play context-dependent roles [38,85].

## 3. RUNX1 and CBFβ in Breast Cancer: The Enigmatic Duo

While RUNX1 and CBFβ have been most widely studied in the context of blood cancers, they have over the last decade been implicated in breast cancers, both in the context of tumor suppression and tumor promotion [38,85,90,91]. In ER+ breast cancer, somatic mutations such as frame-shift mutations and point mutations, and deletions of *RUNX1,* have pointed to a tumor suppressive role [85] (Figure 1). *RUNX1* downregulation contributed to a gene expression signature that predicted poor outcome and metastatic risk in primary tumors [92]. In a study by Ellis et al., utilizing high throughput next generation sequencing and bioinformatics, *RUNX1* was identified as one of 18 recurrently mutated genes in the luminal-B (ER+) subtype of breast cancers [93]. An independent study also reported two *RUNX1* mutations and four mutations in *CBFB* in ER+ luminal tumors [94], while Nik-Zainal and colleagues showed that both *RUNX1* and *CBFB* were in the top 50 of 93 genes at risk of acquiring driver mutations in breast cancers [11]. Downregulation of *RUNX1* was specifically noted in metastatic breast cancers, and attenuated expression of RUNX1 protein was reported in high-grade breast tumors when compared to low or mild-grade tumors [92,95,96]. Deletions and somatic mutations of *RUNX1* have also been associated with poor differentiation of malignant tumors [97]. A causative relationship between knockdown of *RUNX1* and hyper-proliferation in MCF10A cells (immortalized normal mammary epithelial cells) was described and linked to malignant morphogenesis [95]. 

RUNX1 has been shown to facilitate AXIN1 mediated suppression of β-catenin by inhibiting estrogen-dependent signaling, specifically in ER+ disease [98], where low expression of *AXIN1* was associated with low *RUNX1* expression, unveiling a potential mechanism for RUNX1-mediated tumor suppression. RUNX1 has also been shown to exert protection against epithelial-to-mesenchymal transition (EMT) by antagonizing the oncogenic effects of YAP1—a transcription co-factor implicated in breast cancer metastasis [99].

The gain in metastatic potential of cancer cells, including breast cancer, has been associated with loss of E-cadherin, a protein crucial in cell differentiation, maintaining cell polarity, cell-to-cell interactions and thus normal morphology of tissues [100,101,102]. RUNX1, in association with transcription factors EP300 and hepatocyte nuclear factor 3 (HNF3), has been shown to directly bind to regulatory regions of the gene encoding E-cadherin and to promote its transcription [103]. Re-expression of E-cadherin through overexpression of HNF3 induces morphological changes in metastatic breast cancer cell lines to a more epithelial-like phenotype and reduces their invasive, migratory capacity. RUNX1 was shown to have a synergistic effect on HNF3 function and in promoting E-cadherin expression, which explains another mechanism for its tumor suppressor properties [103]. 

Conversely, there is evidence of *RUNX1* over-expression in various cases of ER negative and triple negative breast cancer, supporting a pro-oncogenic role of RUNX1 correlated with poor prognosis and survival [104,105,106,107,108]. *RUNX1* expression was noted to be significantly high in some cases of the more invasive subtypes of breast cancer [96]. Studies analyzing tissue microarrays (TMA) from TN breast cancer patients have correlated elevated RUNX1 expression to poor survival outcome [108,109,110]. In vitro studies using ER negative breast cancer cell lines have shown that RUNX1 associates with super-enhancers known to interact with other oncogenic transcription factors, such as MYC [111]. In mouse-derived mammary tumors, as well as in the MDA-MB-231 TN cell line, loss of RUNX1 was shown to reduce cell proliferation, migration and invasion [112]. This was supported by results using the *MMTV-PyMT* breast cancer mouse model, associating increased *Runx1* expression with tumor progression [110], whilst *Runx1* upregulation was associated with radiation-induced mammary tumors in rats [113]. Underpinning this phenotype, it has been suggested that RUNX1 binds directly to various proteins regulating transcription of genes involved in the control of mammary tumor promotion, such as *FOXP3, GJA1* and *RSPO3* [112]. 

Highlighting the interchangeable role of RUNX1 is a study showing that, when FOXO expression is constant, downregulation of *RUNX1* supports deregulated cell proliferation. However, the same status of RUNX1 in FOXO-deficient cells induces growth arrest [114]. Both these factors were also negatively correlated in TN breast tumor samples examined by Wang et al. [97]. These findings consistently demonstrate how the role of RUNX1 is dependent on the subtype of breast cancer (ER+ vs. ER-) and that the effect of RUNX1 on tumor cells depends on the context under which it is expressed (Figure 3). To this end, further research is required to unravel the mechanisms by which RUNX1 exerts its dualistic phenotypes. Considering how heavily dependent RUNX1 is on CBFβ for not only its function as a transcription factor but also in the regulation of its translation [37], it is not surprising to see that CBFβ shares an equally enigmatic role (as discussed below).

## 4. RUNX2 in Breast Cancer: Mediator of Metastasis

RUNX2 has been shown to antagonize ER signaling (see below) and to play a tumor-suppressive role in MCF7 breast cancer cells and ER-positive breast cancer patients [38,115,116]. However, it is increased expression of *RUNX2* that is generally associated with breast cancer transcriptomic datasets in both ER positive and ER negative subtypes (Figure 1), and accordingly, RUNX2 has been shown to adopt a pro-oncogenic role in breast cancer cells. This evidence has been extensively reviewed elsewhere [117,118,119]. While displaying context-dependent roles in breast cancer (Figure 3), RUNX2 has built up a prominent reputation as a driver of breast cancer metastasis. As a master regulator of osteoblasts and bone development, increased expression of RUNX2 has been implicated in breast cancer metastasis to bone [120,121,122,123]. In metastatic breast cancer cells, phosphorylation of RUNX2 by AKT to enhance RUNX2 mediated transcription has been shown to increase expression of genes involved in tumor cell invasion [124]. A study by Barnes et al. highlighted the importance of RUNX2 in the growth of MDA-MB-231 and LCC15-MB metastatic breast tumor cells within the bone microenvironment. Here, a genetically inactivated RUNX2, which lacked the transactivation domain, eliminated the original potential of the tumor cells to regulate osteogenic differentiation in bone marrow stromal cells in vitro and their ability to induce osteolytic disease in vivo [125]. This was supported by a separate study where RUNX2, in conjunction with CBFβ, mediated the inhibition of osteoblast/osteoclast differentiation through the induction of sclerostin secretion in MDA-MB-231 cells [126]. Sclerostin is known to antagonize the Wnt signaling pathway in osteoblasts and to interfere with bone development, which could contribute to the growth of secondary tumors in the bone [126,127]. Furthermore, several genes involved in the differentiation of osteoclasts and in resorption of bone, such as *CSF2*, *IL8*, *SPP1*, *SPHK1*, *PTHLH* and various Matrix metalloproteinases (MMPs), were shown to be regulated by RUNX2 in both breast and prostate cancer cells [117,121,126,128,129,130]. RUNX2 with CBFβ has been deemed critical for the expression of Osteopontin/IBSP, IL11 and GM-CSF/CFS2 in metastatic breast cancer cells. These factors induce destruction of bone tissue by osteoclasts, thereby allowing breast tumor cells to invade the bone microenvironment [126,131,132]. Therefore, breast cancer cells expressing RUNX2 and CBFβ can exploit the various growth factors involved in the homeostasis of bone formation to allow tumor cell mediated osteoclastogenesis and growth of metastatic cells in this environment. In support of this, RUNX2 expression was positively correlated with an aggressive phenotype of human breast cancers through immunohistochemical analysis of primary tumors [133]. High expression of *RUNX2* has been particularly associated with invasive, ER-negative cell lines resembling the basal subtype of breast cancer, such as MDA-MB-231, HCC38 and MDA-MB-157 [134]. *CBFB* expression was also notably high in most of these cell lines [134]. Furthermore, in a comprehensive TMA of human breast cancers, high expression of RUNX2 was correlated with ER-negative disease and poor prognosis of patients as well as inducing preneoplastic changes in naïve murine mammary epithelial tissue [135]. In vitro experiments using ER+ MCF7 breast cancer cells demonstrated a role of RUNX2 in inducing EMT and metastasis through its target gene *SNAI2* via the Wnt/TGFβ pathways. Interestingly, in this study, high expression of *RUNX2* and *SNAI2* was unusually correlated with high expression of ERα [136]. RUNX2 expression can also drive tumorsphere formation, a marker of stemness and, in partnership with TAZ, can result in increased shedding of soluble E-cadherin, which in turn can cooperate with HER2 in driving breast cancer cell growth [137].

## 5. RUNX3 in Breast Cancer: Putative Tumor Suppressor

*RUNX3*—found to be inactivated in various cancer types, such as bladder, lung, gastric, colon, liver and breast—has been marked as a classic tumor suppressor gene in multiple studies [82,134,138,139,140,141]. This inactivation might be owed to the fact that the genetic location of RUNX3 clusters with a range of tumor suppressors and hyper-methylation of chromatin is common in this region [142,143,144,145]. Indeed, *RUNX3* gene deletion, polymorphisms and mis-localization of protein have been noted in breast cancers [134,146,147]. Hemizygous deletion of *Runx3* in BALB/c mice was enough to induce the spontaneous development of mammary ductal carcinoma with increased Ki67 staining, confirming a hyperproliferative phenotype in the cancer cells [148]. Restoration of *RUNX3* expression through retroviral vectors in MCF7 breast cancer cells, where *RUNX3* is naturally hypermethylated, has been shown to inhibit proliferation and clonogenic potential of the gene in both in vitro and in vivo models [148]. This anti-proliferative phenotype was associated with the ability of RUNX3 to induce proteasomal degradation of the estrogen receptor. RUNX3 expression led to down-regulation of the ER protein and suppression of ER mediated transactivation and cancer cell proliferation in multiple breast cancer cell lines [148]. In support of this, experiments using a xenograft mouse model reported suppression of growth and invasiveness of MDA-MB-231 tumor cells with ectopic expression of RUNX3 [134]. In a more recent study, RUNX3 expression in YAP1-expressing normal mammary epithelial cells, and breast cancer cells, suppressed YAP1-mediated proliferation, migratory capacity and EMT [99]. Mammosphere assays using Hs578T cells, with particularly high levels of YAP1, indicated that RUNX3 significantly compromised the stemness potential in these cells potentially through direct interaction with YAP1. These results were also reflected in patient cohorts where high YAP1 expression depicted significantly better survival prognosis if they also expressed high levels of RUNX3 (as well as RUNX1) [99]. On the other hand, patient tumor samples with high YAP1 but low RUNX1 and RUNX3 depicted higher levels of gene signatures associated with EMT and cancer cell stemness [99]. A similar phenotype was reported in a separate paper where RUNX3 was shown to suppress EMT and stem-cell-like properties exerted by PIM1—an oncogenic kinase known to contribute to breast tumorigenesis [149]. Additionally, RUNX3 was shown to negatively regulate genes associated with infiltration of immune cells into the breast tumor microenvironment and with poor prognosis [150]. While many studies provide evidence for RUNX3 in breast cancer suppression, we note that others have discussed the expression status of this transcription factor in some epithelial lineages [151]. 

Interestingly, gene expression profiling of primary breast cancer stromal cells identified *RUNX3* as part of a 26-gene signature that was correlated to poor clinical outcome [152]. Here, high *RUNX3* expression in tumor stroma was shown to be positively associated with poor prognosis. mRNA upregulation of *RUNX3* has been noted in primary breast tumor samples through whole genome sequencing analysis (Figure 1). It might be hypothesized that this increase in expression in bulk tumors reflects high *RUNX3* expression in the tumor stromal compartment. 

## 6. *Runx* Genes in Mammary Development and Homeostasis

Both RUNX1 and RUNX2 appear to have important roles in normal mammary development, but evidence for a major role for RUNX3 is currently lacking; indeed, some studies have failed to demonstrate meaningful levels of gene expression in the mammary epithelia [97,135,153,154]. Echoing studies in other lineages, the *Runx* genes appear to engage in a delicate interplay in regulating the balance of stem and progenitor cells within the mammary system. Our group previously showed that *Runx2* was expressed at high levels in mammary cells with a stem-like phenotype and was critical for mammosphere and colony formation, indicating an important role in regeneration [155]. In contrast, silencing of *RUNX1* in MCF7 cells was associated with the upregulation of stem cell markers [98]. Using fractioned populations of MCF10AT1 cells, Fritz et al. [153] showed that *RUNX1* is decreased, and *RUNX2* is higher in the population characterized by enhanced stem or progenitor properties. Moreover, *RUNX1* is required for cells to exit a bipotent stem-like state and to drive differentiation and the development of ducts and lobules in MCF10A cells [156]. These results are supported by independent studies, which reported that *RUNX1* expression inhibited tumorsphere formation in breast cancer cell lines [157]. The concept that RUNX1 may drive differentiation was also supported by studies conducted by the Li lab [154]. This group showed that *Runx1* is expressed in both basal and ductal luminal cells but not in alveolar luminal cells of the mouse mammary gland. Genetic deletion of *Runx1* using MMTV driven Cre (which is mainly expressed in the luminal lineage) revealed that the luminal progenitor population was maintained whilst the mature luminal phenotype was reduced, suggesting that *Runx1* is an important player in this differentiation step. Furthermore, loss of *Runx1* significantly reduced the ER+ subpopulation of luminal cells indicating that *Runx1* is required for the development or maintenance of these cells. Like other cancers, blocked differentiation could lead to a relative accumulation of immature cells more susceptible to transformation. Consistent with this, loss of either *Tp53* or *Rb1* tumor suppressors, combined with *Runx1* loss not only rescued the ER+ population but resulted in hyperproliferation and expansion of these cells compared to the loss of *Rb1* or *Tp53* alone [154]. Thus, it can be hypothesized that loss of *Runx1* in the luminal lineage may represent a preneoplastic event susceptible to secondary events. Indeed, deletion of *Runx1* in a stem cell population of luminal cells marked by the RUNX1 intronic enhancer (eR1+) [158] mediated the development of luminal hyperplasia and enhanced mammary organoid growth. 

Expression studies indicate that *Runx2*, like *Runx1*, is expressed in both the luminal and basal mammary epithelium, with levels being higher in the latter [135], and is especially prominent in the terminal end buds of developing ducts [135,159,160]. *Runx2* is expressed in virgin glands and is maintained through to mid-pregnancy but falls in late pregnancy and during lactation, a pattern of expression that is similar to that seen with *Runx1* [160,161]. Enforced expression of *Runx2* in mammary epithelium (via a MMTV-*Runx2* transgene) [135] or murine mammary epithelial HC11 cells [160], induced a block in late-stage differentiation of mammary epithelium, characterized by defective lobuloalveolar differentiation and failure of lactation, consistent with the physiological fall in *Runx2* expression in mid- to late pregnancy. These studies hinted at a pro-oncogenic function for *Runx2* in the mammary compartment. Aged MMTV-*Runx2* transgenic mice developed hyperproliferation, and even a modest incidence of ductal carcinoma *in situ*, whilst the HC11 cell line displayed a more EMT-like phenotype. Paradoxically, deletion of *Runx2* also resulted in perturbed alveolar development during late pregnancy [160]. It is possible that *Runx2* is required for the development of alveolar progenitors, but expression must be attenuated for terminal differentiation and lactation. In agreement with a suspected pro-oncogenic role, loss of *Runx2* slowed tumor growth in a mammary tumor model [160].

## 7. Relationship of RUNX/CBFβ with ER Signaling

Considering how strongly RUNX/CBFβ-mediated effects are correlated with the ER status of breast cancers, it is obvious that a complicated relationship (either direct or indirect, if not both) exists between the ER and the RUNX/CBFβ heterodimeric complex [91]. However, it is important to consider that ER signaling itself is highly pleiotropic. ER signaling holds the capacity to both promote and suppress breast tumorigenesis, depending on various combinations of molecular signals within the mammary cells that help tightly regulate the course of signaling cascades triggered by estrogens [162]. Firstly, the response to 17-β-estradiol (E2) within the mammary cell is dependent on the expression of ER, and the subsequent post-transcriptional and post-translational modifications, in addition to the successful dimerization of two ERs in the nucleus [162]. On top of that, metabolic processing, and binding of E2 to other G-protein coupled receptors and certain isoforms of ER that undertake non-transcriptional modulation—via activation of kinases and phosphatases—affect the ultimate fate of the mammary cells [162]. In some instances, ER-mediated oncogenesis is achieved due to ER-induced activation of cell proliferation and metastasis mediators, such as ETS1, PI3K/AKT, LRP16, VEGFR2 and extracellular matrix proteins. Conversely, ER-E2 signaling may also be beneficial when E2 is linked to breast cancer cell apoptosis [162], blocking of angiogenesis [163] and, in some cases, via E2-mediated blocking of the oncogenic TGFβ pathway.

As stated above, *RUNX1* and *CBFB* loss-of-function mutations are strongly associated with ER+ breast cancer. In this context, it is intriguing that the hormone receptor ERα and RUNX1 physically interact and co-regulate a large number of target genes. This raises the scenario that the ERα-driven program of gene expression can be modulated by RUNX1 and vice versa. Indeed, the presence or absence of E2 profoundly alters the profile of differentially expressed RUNX1 target genes in MCF7 cells [98]. ERα can control gene expression both directly, via the estrogen response elements (ERE), or indirectly by tethering and cooperating with other transcription factors. There is evidence that RUNX1 is a major “tethering” partner, underlining the intimate relationship between these transcription factors in the control of global gene expression patterns [164]. A functional interaction between RUNX2 and ERα has also been reported. E2-bound ERα suppresses RUNX2’s transcriptional activity. This effect was ligand dependent as ERα/RUNX2 interaction was significantly increased in the presence of E2 [116]. The authors went on to demonstrate that there was an inverse correlation between ERα expression and expression of RUNX2 target genes in breast cancer biopsies. Further, the functional significance of the ERα/RUNX2 interaction was elegantly illuminated in a study by Chimge et al. [115]. This work showed that, in general, E2 antagonized RUNX2 regulation of target genes and vice versa, although cooperation was also noted for a smaller set of target genes. Together, these data indicated that RUNX2 and E2 could profoundly modulate each other’s gene expression profiles. In addition, this study also noted that RUNX2 could directly repress ERα expression, confirming an observation in osteoblasts that RUNX2 could bind to the ERα promoter and regulate transcription [165]. Conversely, it has been previously reported that ERα and the related gene *Estrogen Related Receptor α (ERRα)* regulate RUNX2 itself [166]. Cumulatively, these studies indicate that RUNX activity can redirect or reprogram the effects of E2/ERα in mammary cells, with profound implications for tumor development or progression in mammary tissue. Studies on the relationship between *RUNX3* and ERα are more limited, but Huang et al. [148] exogenously expressed *RUNX3* in MCF7 cells and noted reduced proliferation in the presence of E2 and reduced colony formation in soft agar. The transcriptional activity of E2/ERα was reduced in *RUNX3*-expressing cells due to reduced stability and half-life of ERα. 

## 8. CBFβ as an Emerging Regulator in Breast Cancer

CBFβ is crucial for the operation and function of all three RUNX proteins, and it is becoming apparent that it also plays an important role in cancer etiology [167,168,169,170,171,172,173,174], particularly in breast cancer. In a comprehensive targeted sequencing-based study analyzing almost 1000 primary breast cancer samples, *CBFB* was shown to be one of the top 17 recurrently mutated genes [175], whilst a separate whole-genome sequencing study of 560 breast tumors identified *CBFB* as one of the 93 protein-coding genes harboring potential driver mutations [11]. This data is complimented by results from the METABRIC study [24], where *CBFB* was shown to be altered in 14% of the 2433 breast cancer cases investigated (Figure 1), as well as in 13% of primary breast cancer cases in the TCGA Firehose legacy dataset, and in 5% of metastatic breast cancer samples analyzed in the study by Li et al. [22,23,176,177,178]. Indeed, many whole-genome sequencing and transcriptomic studies have revealed how common alterations and mutations in this gene are [11,94,179]. 

Additionally, these results highlighted that *CBFB* undergoes varying alterations depending on the breast cancer subtype in a similar way to that for RUNX1. As highlighted in Figure 1, *CBFB* tends to be most frequently altered in ER-positive breast cancers where truncating or missense mutations and gene deletions are commonplace; whereas it is less frequently altered in ER-negative disease and often found to be upregulated [10,24,85,93,94]. 

### 8.1. CBFβ as a Gate Keeper

In ER-positive breast cancer, the presence of mostly loss-of-function mutations and deletions of *CBFB* suggests that it may be acting to inhibit tumorigenesis in this context [37,175]. The missense mutations noted in *CBFB* are focused around the RUNT-binding domain and therefore would abrogate the interaction between CBFβ and RUNX proteins [175,180]. Emerging evidence has shown that removing *CBFB* in ER-positive MCF7 cells via CRISPR-Cas9-mediated gene deletion increases ER-dependent migration of these cells. The activated RUNX1/CBFβ complex suppresses ER-mediated activation of the mitogen Trefoil factor 1 (TFF1) and thus inhibits migration [180]. CBFβ is also crucial for the inhibitory function of RUNX1 in the ERα mediated repression of AXIN1 [98]. AXIN1 is known to repress the Wnt signaling pathway, and when *CBFB* is deleted in ER-positive cells, this repression of the cell proliferative Wnt pathway is removed [180].

The genetic evidence that *CBFB* loss-of-function mutations are associated with breast cancer raises the question of whether these effects are simply due to loss of RUNX transcriptional function via impaired DNA binding (or reduced protein stability) or whether CBFβ exerts RUNX-independent effects that influence cancer development through an alternative mechanism. Evidence for such a role was provided by Malik et al., [37] who showed that, distinct from its transcriptional partnership with RUNX, CBFβ protein could bind to mRNA and cooperate with the translation initiation factor eIF4B to regulate translation of numerous gene transcripts (>800), including *RUNX1* itself. This study also showed that loss of *CBFB* led to the transformation of MCF10A cells in vitro and tumorigenicity in vivo [37]. The transformation phenotype could be reversed by re-introduction of *CBFB* or the loss of *NOTCH3*, suggesting that the latter was negatively regulated by RUNX1/CBFβ in these cells and was mediating transformation [37].

In a later paper, Malik et al. also identified mutual exclusivity between alterations in *CBFB* and *TP53* through in silico interrogation of patient tumors [181]. Here, a functional relationship was suggested where p53 and the RUNX1/CBFβ complex cooperate to exert their tumor-suppressive roles in normal breast cells through activation of TAp73. Loss of TAp73 facilitated the oncogenic effect of NOTCH3 over-expression in the induction of tumorigenesis. Furthermore, this study identified 32 additional targets regulated by both CBFβ and p53, which could potentially be involved in their anti-tumorigenic functions [181].

### 8.2. CBFβ as a Driver of Tumorigenesis

It is interesting, within the context of the foregoing, that evidence for a pro-tumor role of *CBFB* in breast cancer has also been proposed. According to the METABRIC study, elevated expression of *CBFB* has been observed in 3% of breast cancer patients although, notably, gene amplification and high expression of mRNA were predominately in ER-negative patient samples [22,23,24]. The concept that gain-of-function mutations could contribute to tumor development or an aggressive phenotype is intriguing, as early studies on the RUNX/CBFβ relationship indicated that RUNX levels were tightly controlled whilst CBFβ expression was thought to be in abundance and depended on RUNX for translocation to the nucleus. However, gain-of-function mutations suggest that either, in some contexts, CBFβ is rate limiting, or this effect is independent of the *RUNX* genes (as described above).

High *CBFB* expression has been positively correlated with increased metastasis and poor prognosis of patients [182]. Supportive of the clinical data, the Shore group showed that CBFβ and RUNX2 contributed to the metastatic phenotype of the ER-negative breast cancer cell line MDA-MB-231 [183]. Indicative of their cooperative function, CBFβ bound to RUNX2 was found in the nucleus of metastatic cells and the transcription co-factor was deemed essential for the expression of various genes associated with invasive phenotypes, such as osteoclast-promoting *SPP1, BGLAP, MMP9, MMP13*, *CSF2* and *IL11* and osteoblast-inhibiting *SOST,* which encodes Sclerostin. Invasion assays with knockdown of *CBFB* showed a 90% reduction in the migratory ability of MDA-MB-231 cells, a characteristic subsequently rescued upon re-introduction of *CBFB* [183]. Additionally, CBFβ, in conjunction with RUNX1 and RUNX2, was shown to mediate EMT in MDA-MB-231 cells through regulating expression of *SNAI2*—a known transcription factor involved in EMT [184]. Depletion of CBFβ reversed EMT; suppressed the capacity of these cells to grow in co-cultures with osteoblasts in vitro; and significantly reduced their propensity to metastasize into bone microenvironments in vivo [184]. High expression of CBFβ, and a consequent increase in tumor cell invasiveness and migratory potential, has been demonstrated in two further metastatic breast cancer cell lines. Knockdown of *CBFB* in the metastatic MDA-MB-436 cell line also resulted in reduced tumor growth and improved overall survival in a xenograft model [182]. Migration, invasion, expression of EMT, and bone modulating markers, such as *VIM*, *SNAI1*, *BGLAP*, *CXCR4*, as well as *RUNX2*, were also reduced in response to loss of *CBFB* [182]. These properties allow breast cancer cells to invade the bone microenvironment and modulate bone cells to allow development of secondary tumors. Interestingly, circulating exosomes derived from the serum of breast cancer patients with bone metastasis demonstrated significantly higher levels of *CBFB* compared to those derived from healthy patients or patients with no observable metastasis [182]. These CBFβ-mediated phenotypes seemed transferrable through the exosomes since breast cancer cells with low metastatic potential (T47D and MCF12A), when treated with media containing high *CBFB* expressing exosomes, mimicked their metastatic counterparts, exhibiting increased migratory and invasive properties. Overexpression of *CBFB* in the same cell lines recapitulated the effect with exosome treatment, confirming the oncogenic role played by CBFβ in these cells [182]. Since 70% of metastatic breast cancer patients develop incurable bone metastases [185], finding novel targets such as RUNX2/CBFβ to inhibit or delay this process has the potential to improve disease prognosis and survival of such patients. In fact, a small molecule inhibitor of the RUNX/CBFβ complex (AI-10-104) has been shown to effectively disrupt the CBF complex in breast and ovarian cancer cells [168,186]. Through direct binding to CBFβ and allosterically inhibiting its interaction with RUNX proteins, the inhibitor prevents translocation of the complex into the nucleus, thereby compromising RUNX/CBFβ-mediated transcription [186]. In an in vitro 3D model of basal-like breast cancer, use of the RUNX/CBFβ inhibitor was shown to exert a striking growth-inhibitory effect with almost complete suppression of cell survival and colony formation [186]. An alternative small molecule inhibitor (CADD522) has been shown to block *RUNX2* gene regulation by interfering with RUNX2-DNA binding and could antagonize the growth of breast cancer cells [187]. Although this inhibitor could interfere with all three RUNX proteins, it appeared to have greatest specificity for RUNX2 [187]. These studies open up a potential avenue for treating TN and basal like subtypes of breast cancer. 

## 9. Concluding Remarks

It is evident that the RUNX/CBFβ transcriptional complex has an important role in breast cancer biology (summarized in Figure 3) and in normal mammary epithelial development, not least through its intricate relationship with ER/estrogen signaling. We are at the precipice of our mechanistic understanding of how these proteins contribute to disease progression, which has implications for how this complex might be therapeutically targeted. With inhibitors being actively pursued [186,188], identifying which patients might benefit from disruption of the RUNX/CBFβ signaling cascade, possibly in combination regimes, has important potential. We have specifically concentrated in this review on our knowledge of RUNX/CBFβ within the context of the breast epithelia, but it is exciting that these proteins may also have an important role (direct or indirect) within the tumor microenvironment. As noted above, *RUNX3* was part of a poor prognostic stromal gene signature in breast cancer patients [152], and very recently, RUNX1 (and also RUNX2) was identified through an epigenetic analysis of murine mammary tumors to be highly upregulated in cancer-associated fibroblasts [189]. These authors extrapolated their findings by applying multiplex immunofluorescent staining to show that RUNX1 was expressed in human breast cancer fibroblasts but not in normal breast tissue, while a RUNX1-stromal signature could stratify patients. Furthermore, although only correlative, a recent plethora of papers has proposed that the mutational status of *RUNX* genes impacts the immune cell microenvironment in breast (and other) cancers, with an increased infiltrate associated with high levels of RUNX expression [150,190,191]. Whether this could act as a biomarker for certain patient groups, or holds functional relevance is yet to be explored. As master regulators in both development and disease, the RUNX/CBFβ complex holds multifaceted attributes that need to be unraveled further to understand the mechanisms behind their complex and context dependent role in breast cancer.

## Figures and Tables

**Figure 1 cells-12-00641-f001:**
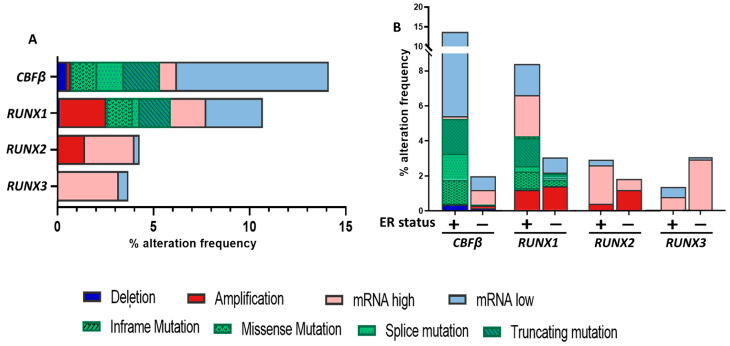
(**A**) Genetic alteration frequencies for each of the 3 *RUNX* genes and *CBFB* in all breast cancer patients. (**B**) Data as in A but separated by ER status of breast tumors to illustrate the different spectrum of *RUNX* and *CBFB* alterations dependent on disease subtype. Data acquired from the METABRIC dataset using cBioportal accessed January 2023 [22,23,24].

**Figure 2 cells-12-00641-f002:**
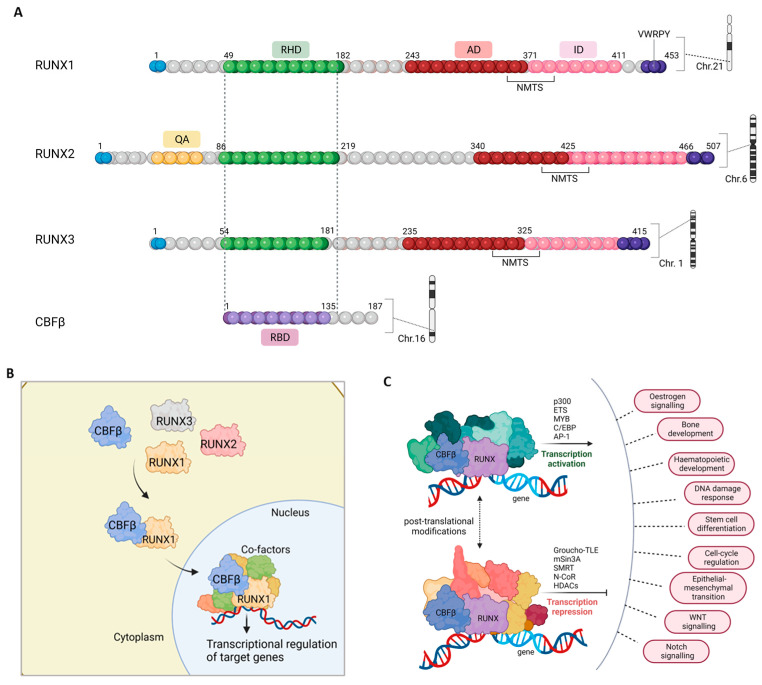
The RUNX/CBFβ complex. (**A**) The human RUNX proteins are encoded by 3 *RUNX* genes, *RUNX1* (on chromosome 21), *RUNX2* (on chromosome 6) and *RUNX3* (on chromosome 1), while CBFβ is encoded by the *CBFB* gene on chromosome 16. The Runt-homology domain (RHD) on RUNX proteins binds to the Runt-binding domain (RBD) on CBFβ to allow formation of the CBF complex. The activation domain (AD), inhibitory domain (ID) and nuclear matrix targeting signal (NMTS) are indicated. The QA region on RUNX2 is an extended region of glutamine-alanine repeats that differentiates it from the two other RUNX family members. The carboxy-terminal VWRPY motif is used in the interaction with co-factors. (**B**) CBFβ interacts with members of the RUNX family within the cytoplasm. Once bound to any of the three RUNX proteins, in this case RUNX1 is depicted, the complex is translocated into the nucleus where it can bind to DNA and regulate transcription. Recruitment of various co-factors determines the fate of transcriptional regulation of RUNX/CBFβ target genes. (**C**) The core binding factor complex works in conjunction with an array of transcription co-activators and co-repressors to regulate crucial cellular pathways. Figures created using Biorender.com and taken from the PhD thesis of Adiba Khan, University of Glasgow 2022.

**Figure 3 cells-12-00641-f003:**
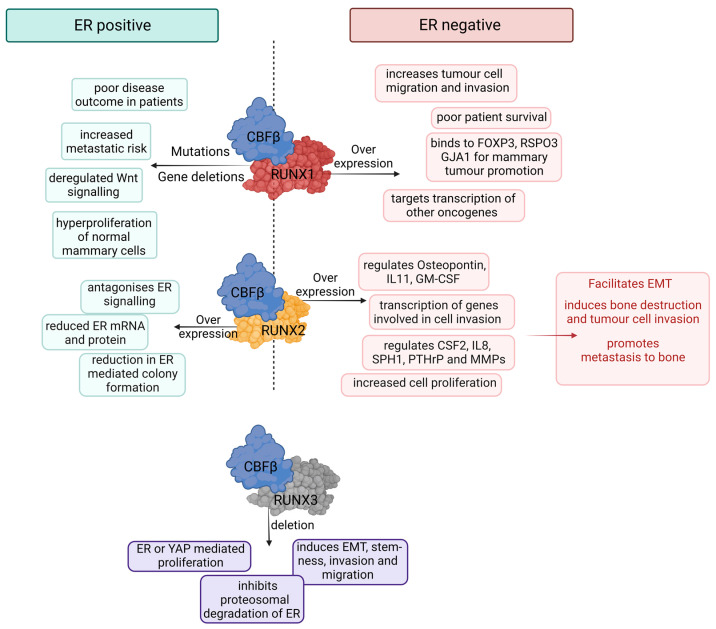
Schematic summarizing the knowledge around each of the RUNX proteins in ER positive and ER negative breast cancer. The RUNX1/CBFβ complex acts in a tumor suppressor manner in ER positive disease and is pro-tumorigenic in ER negative disease. The RUNX2/CBFβ complex is a critical regulator of metastasis in ER negative disease and has been shown to antagonize ER/E2 signaling inducing reduced expression of ER mRNA and protein. *RUNX3* hypermethylation, gene deletions and polymorphisms have been reported in breast cancer samples while gene deletion in vitro is associated with ER and YAP1 mediated proliferation, induction of EMT, invasion and migration of breast cancer cells.

## Data Availability

Not applicable.

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
