# Peer review of "The RUNX/CBFβ Complex in Breast Cancer: A Conundrum of Context"

_cells, 2023, doi:10.3390/cells12040641_

Round 1

Reviewer 1 Report

The review manuscript by Khan et. al., summarizes recent findings related to RUNX/CBFb complex in breast cancer. This is a nice and comprehensive review that well covers important aspects in this topic. Also, this review fairly discusses about oncogenic and tumor suppressive roles of RUNX/CBFb complex and its context dependent function. I have just a few comments.   

Comments.

1.     References after #159 are not listed. Please add them.

2.     I am not sure reference #158 (Thesis in University of Glasgow) is easily available. If not, this is not used for reference.

3.     EMT in line 178 is better to be spelled out. 

4.     In line 183, more explanation on HNFs helps readers to understand the content.

5.     In line 334, explanation of what is eR1+ is necessary. 

6.     In line 357, put RUNX/CBFb before heterodimer. 

7.     In line 490, comma after MET is not necessary.

8.     In line 521, place reference showing that the inhibitor has greater specificity for RUNX2.

9.     Please edit English for easier reading.

Author Response

We thank the reviewer for their kind comments and helpful suggestions which we believe improves our manuscript. All suggestions have been changed.

Please see attachment for point-by-point response.

Reviewer 2 Report

SIGNIFICANCE

This review provides a very informative and comprehensive overview of the role of RUNX proteins (and its partner CBFB) in breast cancer. This is a very active research area because >16,000 papers on RUNX proteins have been published, and >5,000 of these have focused on the role of RUNX proteins in cancer. The authors have done an outstanding job at selecting relevant and impactful papers (>150 references), including those from by the Blyth & Cameron group itself who have been leading experts in the field and have deciphered major roles for RUNX proteins in mammary gland development and breast epithelial cell differentiation.  

This review is quite informative and very well received. Figure 3 provides a beautiful overview of RUNX function in ER+ vs ER- breast cancer. The following recommendations could be considered to improve this excellent manuscript.

SPECIFIC POINTS

1) The papers discusses RUNX1, RUNX2 and RUNX3 mostly as separate entities with many similarities. Yet, these proteins are autoregulated and cross-regulated. The authors may wish to introduce this concept in one strategically positioned sentence somewhere.

2) “Essentially a genetic disease arising from alterations in the expression pattern of normal genes, the accumulation of multiple mutations involving tumour suppressor genes and oncogenes drives transformation of normal breast cells towards malignancy[9].”

Unclear sentence. Simplify as follows:

Breast cancer is in part a genetic disease in which alterations in gene expression patterns and the  accumulation of mutations in both tumour suppressor genes and oncogenes drives transformation of normal breast cells towards malignancy[9].

3)Typo: “polyoma enhancer-binding protein 2 α (PEP2α)” = PEBP2α

4) Source reference not found: “This region was found to support two functions: allowing the RUNX proteins to bind to DNA and enabling heterodimerisation to CBFβ[41] (Error! Reference source not found.).”

5) “The RHD recognises and binds the 5'-TG(T/C)GGT-3' consensus sequence[52] – or what seems more frequently bound in putative RUNX target promoters, the 5’-R/TAACCRCA-3’ sequence[39, 51].

This sentence will create confusion because it does not acknowledge that GYGG and CCRC are complementary sequences. The site is there but just in different orientation.  

Suggested: The RHD recognises and binds the 5'-TGYGGT-3' consensus sequence[52] – or its inverse complement 5’-R/TAACCRCA-3’ - in putative RUNX target promoters[39, 51](Y=C or T, R= G or A).

6) “acetylation (by p300 and BMP-2)”

p300 is a histone acetyl transferases but BMP2 is not. Gene symbols EP300 & BMP2

Suggested: acetylation (by p300/EP300 and/or in response to BMP2 signaling)

7) “chromatin modelling complex SWI/SNF[69, 70]” = chromatin modelling complex SWI/SNF[69, 70] that is formed by SMARC proteins (e.g., SMARCA1/SNF2L1, SMARCA4/BRG1).

8) “expression of Osteopontin, IL-11 and GM-CSF” = expression of Osteopontin/IBSP, IL11 and GM-CSF/CSF2

9) Section”Runx genes in mammary development and homeostasis”- In this section, gene symbols for RUNX proteins expressed in human cell lines and tumors should be shown in all caps (too many examples to list here).

10) Throughout the paper, dashes in gene names should be avoided. Gene symbols never have dashes.

11) “CBFβ as a tumour suppressor” & “CBFβ as an oncogenic driver”. The terms oncogene and tumor suppressor are highly reserved terms by some classical cancer definitions. It would be safer to refer to these RUNX and CBFβ proteins as bifunctional cancer related proteins that can either promote or inhibit tumorigenesis and/or metastasis depending on the biological context.

12) Throughout the paper, use the historical trivial name of a protein and its HUGO approved protein/gene symbol.

OPN, OC, MMP9, MMP13, CSF-2 and IL-11” = “osteopontin/SPP1, osteocalcin/BGLAP, MMP9, MMP13, GM-CSF/CSF2 and IL11”.

Author Response

We thank the reviewer for their very kind comments and excellent suggestions which have all been addressed and which has improved our manuscript.

Please see attached for point by point responses.
